# Dispersion relation of the collective excitations in a resonantly driven polariton fluid

Petr Stepanov[1], Ivan Amelio[2], Jean-Guy Rousset[1,3], Jacqueline Bloch[4], Aristide Lemaître [4], Alberto Amo[5], Anna Minguzzi[6], Iacopo Carusotto[2] & Maxime Richard [1]

Exciton-polaritons in semiconductor microcavities constitute the archetypal realization of a quantum fluid of light. Under coherent optical drive, remarkable effects such as superfluidity, dark solitons or the nucleation of vortices have been observed, and can be all understood as specific manifestations of the condensate collective excitations. In this work, we perform a Brillouin scattering experiment to measure their dispersion relation $\omega(\mathbf{k})$ directly. The results, such as a speed of sound which is apparently twice too low, cannot be explained upon considering the polariton condensate alone. In a combined theoretical and experimental analysis, we demonstrate that the presence of an excitonic reservoir alongside the polariton condensate has a dramatic influence on the characteristics of the quantum fluid, and explains our measurement quantitatively. This work clarifies the role of such a reservoir in polariton quantum hydrodynamics. It also provides an unambiguous tool to determine the condensate-to-reservoir fraction in the quantum fluid, and sets an accurate framework to approach ideas for polariton-based quantum-optical applications.

[1] Univ. Grenoble Alpes, CNRS, Grenoble INP, Institut Néel, 38000 Grenoble, France. [2] INO-CNR BEC Center and Dipartimento di Fisica, Universita` di Trento, 38123 Povo, Italy. [3] Institute of Experimental Physics, Faculty of Physics, University of Warsaw, ulica Pasteura 5, PL-02-093 Warsaw, Poland. [4] Centre de Nanosciences et de Nanotechnologies, CNRS, Université Paris-Sud, Université Paris-Saclay, C2N Marcoussis, F-91460 Marcoussis, France. [5] Univ. Lille, CNRS, Physique des Lasers Atomes et Molécules, F-59000 Lille, France. [6] Univ. Grenoble Alpes, CNRS, LPMMC, 38000 Grenoble, France. Correspondence and requests for materials should be addressed to M.R. (email: maxime.richard@neel.cnrs.fr)

Upon quieting down the thermal fluctuations in a liquid or a gaseous many-body system by deeply cooling it, and if it does not turn solid, a radical transformation occurs as the system behavior starts to be dominated by quantum mechanics. In the case of integer spin particles, a so-called Bose–Einstein condensate appears below a critical temperature, in which a significant fraction of the fluid occupies a single-quantum state[1,2]. The system is then governed by the laws of quantum hydrodynamics, in which the condensate phase $\phi(\mathbf{r}, t)$ plays a central role, as well as the presence of two-body interaction. This framework explains key phenomena, such as irrotationality of the flow, quantization of vortex circulation, coupled amplitude-phase solitons, as well as the occurrence of a superfluid state. Among the most famous examples of such quantum fluids is superfluid helium[3,4], in which this regime has been pioneered, and ultracold atom gases[5] that are nowadays among the most accurate systems to investigate every subtleties of quantum fluids.

The root cause of these fascinating phenomena can be traced back to the nature and dispersion relation (DR) of the elementary excitations in the quantum fluid. As a general feature, these excitations consist of coupled phase and density fluctuations, and due to two-body interaction they are collective in nature. In the mean-field regime, where the temperature is low, interactions are weak, and the condensate fraction is dominant, Bogoliubov derived in 1947 an analytical description of this regime, and could thus reveal the link between the DR of the excitations and the superfluid state[6]. The corresponding experimental measurements of the elementary excitations came decades later in an ultracold atom gas[7–9] and confirmed their key role in the observed phenomena.

Recently, quantum fluids of light have emerged as a new class of quantum fluids, characterized by their nonequilibrium character[10,11]. A paradigmatic member of this class is the fluid of exciton–polaritons (polaritons), that can be pumped within the spacer of an optical microcavity in the strong coupling regime[12]. Polaritons can be understood as photons dressed by semiconductor electrons–hole pairs (excitons) that display significant binary interactions mediated by Coulomb interaction. Their nonequilibrium character comes from the fact that they need to be continuously replenished by an external pump to compensate for ultrafast radiative losses. They thus require a different theoretical framework than their equilibrium counterpart, which is based on the generalized Gross–Pitaevskii equation (GGPE)[13].

In this nonequilibrium setting, the notion of superfluidity raises intriguing questions about its definition and characteristic observables[14–19]; experimentally, a superfluid-like frictionless flow has been demonstrated in 2009 in a steady-state polariton fluid[20]. From there on, a number of quantum hydrodynamics phenomena have been studied in this system, including the nucleation of topological excitations, such as solitons[21,22] and quantized vortices[23–25], phenomena associated with their spin degree of freedom[26,27], vortex dynamics[28,29], and turbulence effects[30].

As mentioned above, the DR of the collective excitations play a crucial role in these phenomena. To the best of our knowledge, while it has been investigated in some particular cases, its measurement under resonant optical drive is still missing. In ref. [31] for instance, a transient polariton population was created with a resonant pulsed laser, and the time-integrated DR of the excitations was measured in a four-wave mixing arrangement, hinting at sound-like collective excitations. A measurement under resonant continuous wave (CW) drive has been carried out in ref. [32], where two bands of positive and negative frequencies excitations have been measured, and found in agreement with the normal and ghost branches of Bogoliubov's theory. However, the optical pump was kept too low to observe collective features beyond the perturbative regime of the single-particle picture. The DR of collective excitations has also been measured in the regime of pulsed nonresonant optical drive in several works[33–35], but in this incoherent excitation configuration, the nature and DR of the excitations are different from those realized under resonant drive condition like ours[36]. Moreover, the pulsed excitation introduces time-dependent densities, which results in inhomogeneous broadening of the excitation spectrum, and makes comparison with theory harder. Finally, a measurement of the Bogoliubov DR has been reported recently in an equilibrium-like fluid of light in the conceptually different case of a propagating geometry in an atomic vapor cell[37].

In this work, we resonantly drive a nonequilibrium condensate of exciton–polaritons with a CW laser, in which the long-range coherence is directly imprinted by the laser, and not the result of a condensation mechanism. We focus our attention on the high-density regime, in which the interaction energy is comparable or exceed the linewidth, such that the excitations are of collective nature. We perform a direct measurement of their DR using an angle-resolved spectroscopy technique inspired from Brillouin scattering experiments. We find that the results differ strongly from the pure polariton condensate situation described by the GGPE, like for example a speed of sound which is apparently twice too low. Inspired by previous work suggesting that a reservoir of long-lived excitons coexists with polaritons even in this resonant excitation regime[38–43], we developed a theoretical framework in which polaritons can be converted into reservoir excitons (cf. illustration in Fig. 1a, b), and in which the reservoir provides an additional two-body interaction channel. The resulting excitations are of hybrid reservoir density and Bogoliubov-excitations nature, and agree quantitatively with our measurements. While some qualitative analogy may be found with second sound in liquid helium[2], there are major differences due to the nonequilibrium nature of the quantum fluid; moreover, this hybrid nature results in quantitative corrections with respect to the GGPE description, of importance for both past and future works on polaritons quantum hydrodynamics.

## Results

**Experiment**. The experiment is carried out with a liquid helium cooled planar GaAs/AlAs microcavity in the strong coupling regime. The coherent polariton fluid (referred to thereafter as "the condensate") is excited resonantly with a tunable single longitudinal mode CW laser (cf. Methods). A sizable population of excitations is spontaneously created on top of the resonantly driven polariton condensate by the interaction of polaritons with the thermal bath of acoustic phonons naturally present within the solid-state microcavity (see Supplementary Note 3). The polaritons involved in these excitations can then relax radiatively, so that their energy and momentum with respect to the condensate constitute a direct measurement of the DR of the excitations. This emission, that we will refer to as excitations photoluminescence (EPL), is collected by the detection scheme illustrated in Fig. 1c.

The collective excitations that differ from free-particle excitations live within a small frequency window surrounding the condensate, of width comparable with the interaction frequency $gn \gtrsim 0.5$ meV/$\hbar$, where $g$ and $n$ are the polariton–polariton interaction constant and the condensate density, respectively. We thus isolate the EPL $I_e$ from the much brighter condensate intensity $I_0$, and from the Rayleigh scattered laser light using a two-stage filtering scheme: in the first stage, we profit from the fact that the condensate has a nonzero cross-polarized component caused by a weak residual birefringence and a weak TE-TM splitting[44], to detect the EPL in the cross-polarized direction with respect to the laser. For the second stage, we designed an narrow

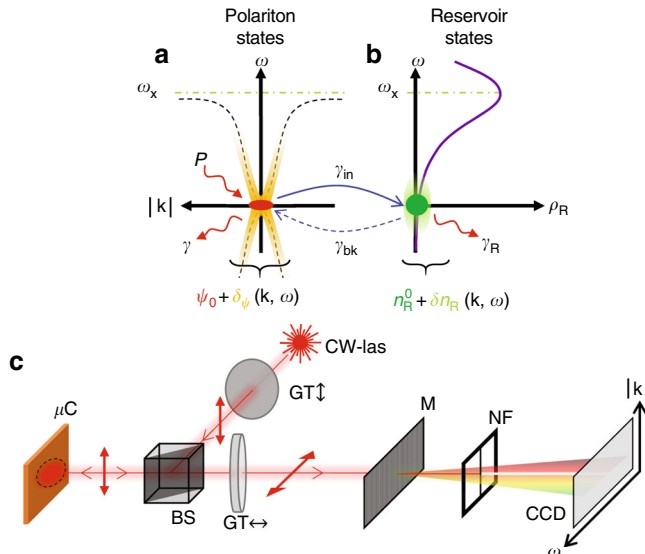

**Fig. 1** Illustration of the different fluid components and of the detection scheme. **a** The polariton condensate (red spot) of wavefunction $\psi_0$, and of radiative loss rate $\gamma_c$ is resonantly excited by the laser of power P. An illustration of the condensate excitations $\delta\psi(\mathbf{k},\omega)$ is shown in yellow in the dispersion plane, with its typical dispersion relation (DR) $\omega(|\mathbf{k}|)$ shown in a black dashed line. The bare quantum well excitonic transition energy $\hbar\omega_X$ is shown as a green dashed line. An illustration of the typical quantum well excitonic density of state $\rho_R(\omega)$ is shown as a purple line in **b**. Owing to their effective mass differences, $\rho_R$'s peak value exceeds the polaritonic density-of-state by 5 orders of magnitude. The low-energy tail of $\rho_R(\omega)$ originates from disorder in the quantum well, and can accommodate a reservoir (green spot) of long-lived excitons (loss rate $\gamma_R$, fluctuations $\delta n_R$ represented in light green). Interconversion of polaritons into reservoir excitons and back by optical absorption or scattering, occur at rates $\gamma_{in}$ and $\gamma_{bk}$, respectively. **c** Sketch of the experimental setup used to measure the DR. The excitation laser light is linearly polarized by a Glan–Thompson polarizer (GT) and passed through a beam splitter (BS) to excite resonantly the polariton fluid. The cross-polarized reflected signal is selected by another GT and passed through a monochromator (M). The polariton emission at the laser frequency is further rejected by a metallic filter playing the role of notch filter (NF), and the remaining EPL is detected on a CCD camera. Some optical elements are omitted, that provide resolution on the EPL emission angle $\theta$, and thus on its in-plane wavevector $|\mathbf{k}| = (\omega/c)\sin(\theta)$ ($\hbar\omega$ is the photon energy, and $c$ the speed of light in vacuum)

band notch filter (labeled "NF" in Fig. 1c) made up of a featureless metallic stripe placed in the output focal plane of the monochromator. The resulting rejection is such that the EPL signal can be well identified even as close as 0.1 meV from the condensate. Figure 2c–e shows measured angle-resolved EPL patterns $I_e(\theta,\omega)$ obtained with this method.

In order to measure sharply defined dispersion relations, we chose two regions of the microcavity of ~50 μm diameter characterized by a weak disorder amplitude of the potential experienced by polaritons $V(x,y)$. They are labeled further on as "working points" (WPs) A and B (For the sake of generality, a third working point is presented and fully analyzed in Supplementary Note 4). The microreflectivity measurement shown in Fig. 2b provides a cross-section of $V$ across WPA showing that its spatial fluctuations are smooth and small as compared to the linewidth. The reference free-particle DR at WPA is extracted from the EPL measurement shown in Fig. 2c. It is obtained under weak excitation at normal incidence, with the laser energy $\hbar\omega_l$ red-detuned from the $\mathbf{k} = 0$ lower polariton

resonance by $\Delta = \hbar\omega_l - \hbar\omega(0) = -0.5$ meV. The corresponding extracted free-particle DR is labeled "0" in Fig. 3a.

We then shifted the laser to $\Delta = 0.79$ meV on the blue side of the polariton resonance, in order to access the regime for which $n(P)$ the condensate density dependence on the driving laser power $P$ exhibits a bistable behavior[45,46]. In the context of the GGPE theory, the regime of collective excitations corresponds to the upper branch of the bistable $n(P)$: at the lower laser power edge of this branch (just before switch down) sits the gapless sonic regime, in which the excitations are expected to be phonon-like with a well-defined speed of sound[11]. Higher up along this branch, a gap opens up for increasing $P$ and the DR adopts a more curved shape. In order to characterize this bistability curve, the unfiltered condensate emission $I_0$ is collected in the cross-polarized direction vs the laser power $P$. Note that in this measurement the excitations have a negligible contribution. The measured $I_0(P)$ curve is shown in Fig. 2a: the lower and upper branch are separated by a sharp jump: indeed since we chop the laser with a 5% duty cycle to prevent unwanted heating effects, the bistable region appears closed. Note that in spite of this technical constraint, we can still get very close to the sonic regime in the measurements, although we can never strictly reach it.

Based on this preliminary calibration measurement, we proceeded to the extraction of the DR of the collective excitations for several laser powers $P$ along the upper branch $I_0(P)$. Due to the stringent requirements of the driving laser beam shape both in Fourier and real space, we chose to work with a large laser spot of Gaussian intensity profile of 50 μm diameter (cf. Methods). As a result, for states in the upper branch of $I_0(P)$, the polariton density is organized into two large spatial structures: The high-polariton density is contained in a large diameter disk-shaped area at the center of the laser spot, separated from a low-density outer region by a sharp switching front (see examples in Supplementary Fig. 5). The nonlinearity thus acts as an effective "top-hat" spatial filter for the Gaussian pump mode, that homogenize the high-density region we are interested in (see details in Supplementary Note 5). We also checked in a lineshape analysis that the influence of the in-plane disorder visible in Fig. 2b is weak and negligible as compared with the features of interest in the dispersion relation (cf. Supplementary Note 5).

In order to collect EPL only from the high-density area, we rejected the outer region using an iris of diameter $D_i$ matching that of the switching front of typically ~35 μm diameter. While this spatial selection introduces a spurious angular spread ≤1.5° to the EPL, it does not prevent resolving the collective features in the dispersion relation, that are visible within a ~5° window as can be seen in Fig. 3. As discussed in Supplementary Note 8, we took this finite angular resolution into account in the simulations. Note that under nonresonant incoherent excitation, another source of momentum broadening would appear due to density-induced radial flow of polaritons[47,48]. This cannot occur in our resonant excitation configuration since the condensate phase is locked by the coherent pump.

Two raw results of angle- and energy-resolved EPL are shown in Fig. 2d and Fig. 2e that correspond to two states on the curve $I_0(P)$ labeled "1" and "2", respectively in Fig. 2a: "1" is a state on the lower branch of $I_0(P)$, while "2" is on the upper branch and as close as possible to the switch-down point.

The key feature we want to focus on in this work is the shape of the collective excitations DR $\hbar\omega_e(\theta)$ (where $\hbar\omega_e = \hbar\omega - \hbar\omega_l$, and $\hbar\omega$ is the detected photon energy), and how it compares with common assumptions. We thus performed a numerical analysis of the raw angle- and energy-resolved EPL measurements in order to determine as accurately as possible the measured DR, with a special care taken on determining the statistical confidence

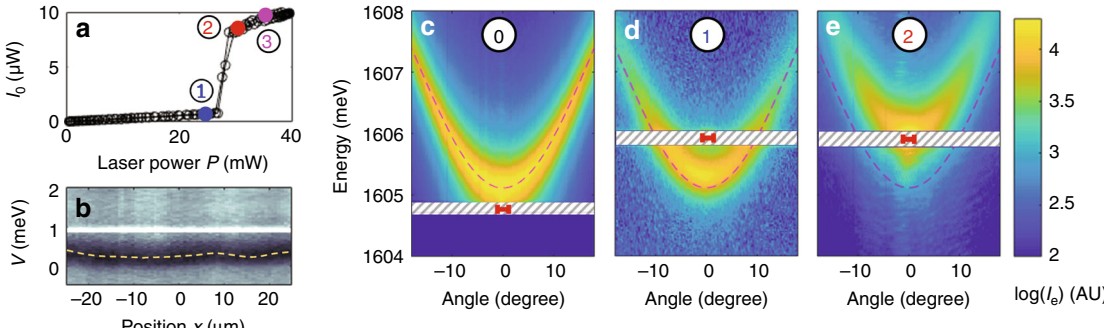

**Fig. 2** Characterization and resonant photoluminescence of WPA. **a** Measured cross-polarized polaritonic emission intensity $I_0$ vs laser power $P$. **b** Color-scaled microreflectivity measured across the working poin. The black area is the polariton absorption dip. The white streak is due to residual laser light. The orange dashed line shows the extracted potential $V(x, y)$ across the working point. Angle and energy-resolved cross-polarized EPL measurements $I_e(\theta_p, \hbar\omega)$ in the free-particle regime (**c**), in the lower branch of $I_0(P)$ (**d**), in the upper branch (**e**). The intensity is color coded on a logarithmic scale. The hatched rectangles show the spectral range rejected by the notch filter. The laser energy and angular spread are shown as a red segment. WPA is characterized by $\delta = +1.2\ 5\ \mathrm{meV}$

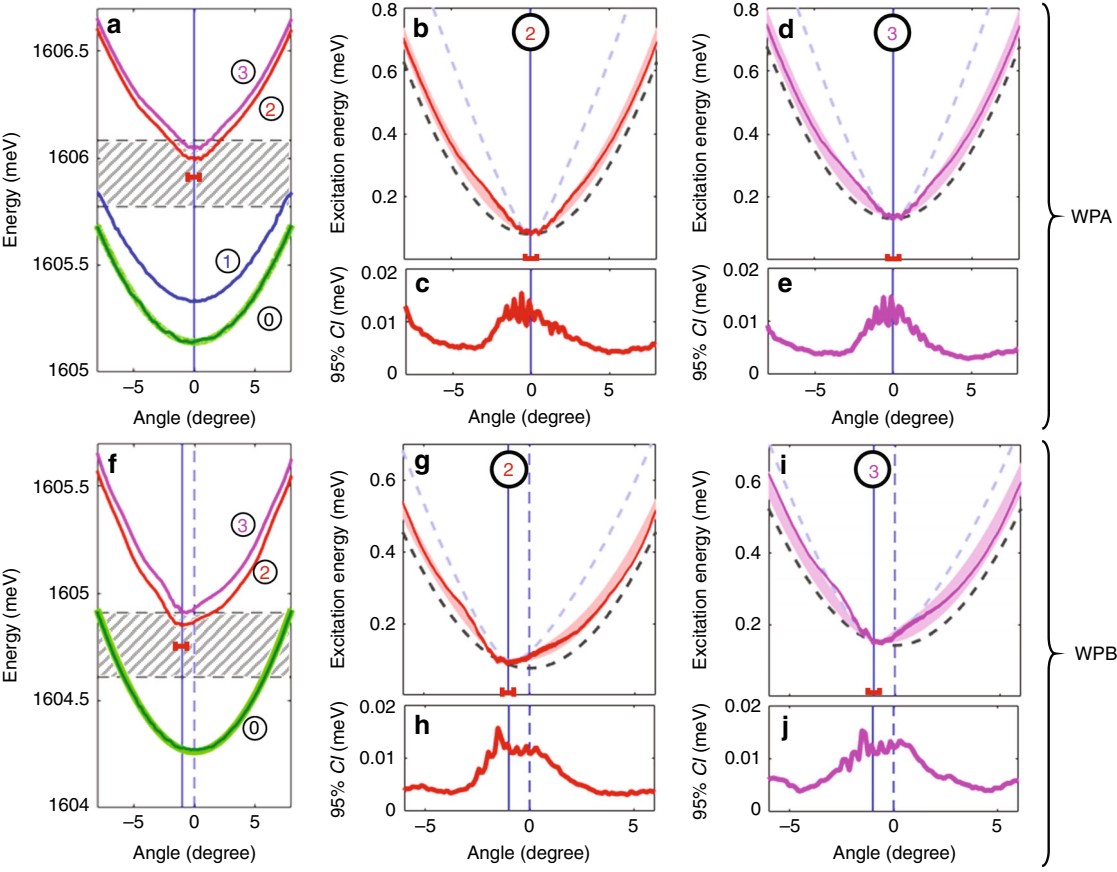

**Fig. 3** Excitation dispersion relations. Measured DR $\hbar\omega(\theta)$ as obtained from the numerical analysis of the EPL (solid colored lines) for WPA (**a**) and WPB (**f**). Four states of the fluids are shown according to their position along $I_0(P)$ (plotted in Fig. 2a for WPA): "0" is the free-particle DR (green line), "1" is a lower branch state of $I_0(P)$ (blue), "2" and "3" are two upper branch states (red and magenta). The calculated free polariton DR is the thick light green line underneath the measured one. **b**, **d** and **g**, **i** are zoomed in plots of DR "2" (narrow red line) and "3" (narrow magenta line) of WPA and WPB, respectively, in units of condensate excitation energy $\hbar\omega_e = \hbar\omega - \hbar\omega_l$. The 95% confidence interval amplitude (CI) of the measured DRs is shown for each DRs in panels **c**, **h** and **e**, **j** for WPA and WPB, respectively, with the same color codes. The theoretical DRs in the rigid blueshift limit (dashed black line) and in the GGPE limit (blue dashed line) are shown in **b**, **d**, **g**, and **i**. The full vectorial theory is also shown as a thick red and magenta lines visible underneath the measured DR; the line thickness represents the 95% confidence interval of the fitting procedure with the data. In each panels, the laser energy and angular spread are shown as a red segment, and the vertical blue line shows the angle of excitation. The hatched rectangle is the spectral range rejected by the notch filter in **a** and **f**

interval of the result (see the Methods section). Figure 3a shows the resulting DRs for WPA, for the free polariton dispersion (dark green line, labeled "0"), and three different blueshifted states (blue, red, and magenta lines, labeled "1" to "3"), with a detailed view of "2" and "3" shown in Fig. 3b and Fig. 3d, respectively, and the 95% confidence interval for the determination of $\hbar\omega_e$ shown in Fig. 3c and Fig. 3e, respectively. The same analysis is shown for WPB in Fig. 3f–j (see Supplementary Note 2 for details on the method). The free polariton DRs in WPA and WPB are very well-fitted with the near-parabolic theoretical free polariton DR (light-green line) as obtained from the coupled oscillators model, and can thus be trusted for comparison purpose.

We first focus on WPA for which the laser drive is at normal incidence: Curve "1" shows the DR of a polariton fluid for which the blueshift with respect to DR "0" amounts to $\hbar\omega_{BS} = 0.18$ meV, and which is still on the lower branch of $I_0(P)$. Its shape is identical to "0", indicating that in spite of the blueshift, the condensate excitations are only weakly perturbed from the free-particle picture. Curves "2" and "3" are obtained in the upper branch of $I_0(P)$ and feature a clearly modified shape with respect to "0", which is an unambiguous signature that the nature of the condensate excitations have changed from free-particle to collective as a result of strong interaction energy ("2" and "3" are blueshifted by $\hbar\omega_{BS} = 0.85$ meV and $\hbar\omega_{BS} = 0.90$ meV, respectively, with respect to "0"). In Fig. 3b and g, the low-energy low-angle part of both DR are compared with the theoretical shape expected in two limiting situations: (i) the condensate density is small as compared with the reservoir density, such that the DR consists of a rigidly blueshifted free-particle dispersion; (ii) the system consists of a pure condensate, without any reservoir fraction so that the DR has the form given by the GGPE.

In mathematical form, the DR in case (i) reads

$$\omega_{RB}(\mathbf{k}) = \omega_0(\mathbf{k}) + g_R n_R - i\,\frac{\omega_c}{2}, \tag{1}$$

where $\omega_c$ is the polariton radiative loss rate, $\hbar\omega_0(\mathbf{k})$ is known from the DR measurement at point "0", and $n_R$ and $g_R$ are the reservoir particles density and their interaction constant with the condensate, respectively. $\hbar\omega_{RB}(\theta)$ is plotted as a black dashed line in Fig. 3b and g. In this model, the interaction term is fixed directly by the blueshift as $\hbar g_R n_R = \hbar\omega_{BS}$. The comparison between this model and the measured dispersion "2" and "3" show a clear mismatch, in which the measured dispersion is steeper. The theoretical shape of the dispersion in case (ii) can be obtained by linearizing the GGPE[13] around the pure condensate steady-state density, which, for a condensate with zero momentum results in:

$$\omega_{bog}(\mathbf{k}) = \omega_p \pm \sqrt{(\omega_0(\mathbf{k}) + 2gn - \omega_p)^2 - (gn)^2} - i\,\frac{\gamma_c}{2}. \tag{2}$$

Like in the previous case, the interaction energy $\hbar gn$, where $g$ is the interaction constant and $n$ the polariton density, is inferred unambiguously from the experimentally measured blueshift $\hbar gn = \hbar\omega_{BS}$, which fixes also the position on the theoretical curve $I_0(P)$. These calculated DRs are shown in Fig. 3b and g as dashed light-blue lines. This time, both for "2" and "3", the measured dispersions are now clearly not steep enough to match the theory.

We then performed the same analysis for the other working point WPB, in which we used a nonzero laser incidence angle $\theta_p = -1°$. In WPB, owing the microcavity tuning, the interactions are smaller by a factor ∼2 and a smaller laser detuning of $\Delta = 0.47$ meV is chosen accordingly. The local disorder is obviously different, but of similar average amplitude and characteristic length, as in WPA. The measured DRs for WPB

are shown in Fig. 3f–j with the same labeling conventions as for WPA. For dispersions "2" and "3", situated on the upper branch of $I_0(P)$, with "2" as close as possible to the switch-down point, an asymmetric shape of the DR is obtained, as expected for the collective excitations when the condensate is subject to a flow of nonzero velocity.

Like in WPA, the comparison with the two theoretical limiting cases are shown in Fig. 3d, i and demonstrates that the measured DRs do not agree with either of them. Our analysis also gives us access to the spectral linewidth $\hbar\gamma$ of the excitations as a function of $k$. The latter is found to be essentially fixed by the radiative linewidth, plus an additional contribution coming from the propagation time within the finite size polariton fluid. A more detailed discussion can be found in Supplementary Note 5.

**Theory and interpretation**. These experimental observations pinpoint the necessity of building a theoretical description of the elementary excitations that includes a contribution from the reservoir. We first check the existence of the reservoir alongside polaritons by exploiting their very different respective lifetimes: we performed a time-resolved polariton photoluminescence decay measurement under pulsed resonant excitation and found that the polariton population exhibits a fast decay component fixed by the polariton lifetime, and a slow decay time component of 400 ps that indicates clearly the formation of a reservoir of long-lived excitons fed by the polaritons. Our best guess considering the experimental indications at hand, is that these low energy excitonic states are due to disorder in the quantum well[49]. A detailed discussion can be found in Supplementary Note 1.

We thus developed a theoretical model with the GGPE for the condensate wavefunction $\psi$ as a starting point, coupled to a rate equation that describes the reservoir density dynamics $n_R(t)$[36]. In addition, due to the cross-polarized measurement technique, this theory needs to account for the polariton polarization degree of freedom. This vectorial theory is presented in details in Supplementary Note 8. For the sake of simplicity, we present thereafter a scalar version in which the coupled equations read

$$i\hbar\partial_t\psi = \left[\hbar\omega_0 - \frac{\hbar^2}{2m}\nabla^2 + \hbar g|\psi|^2 + \hbar g_R n_R - i\frac{\hbar(\gamma_c + \gamma_{in})}{2}\right]\psi + F(t) \tag{3}$$

$$\partial_t n_R = -\gamma_R\, n_R + \gamma_{in}|\psi|^2, \tag{4}$$

where the condensed polaritons of effective mass $m$ are resonantly and coherently driven by a homogeneous pump $F(t)$, their finite lifetime is limited by the radiative loss rate $\gamma_c$, and their (typically much slower) capture rate by the reservoir is $\gamma_{in}$. The polariton–polariton interaction energy is proportional to the density $n = |\psi|^2$ with a coupling constant $g$, while the interactions between polaritons and the reservoir population contribute in an additional interaction energy $\hbar g_R n_R$. The reservoir population lifetime is fixed by the time-resolved experiment to 400 ps, such that $\hbar\gamma_R = 1.6$ µeV $\ll \hbar\gamma_c$. As illustrated in Fig. 1a, b and explained in Supplementary Note 1, the reservoir decay also involves processes in which the reservoir excitons are converted back into noncondensed polaritons. In Eq. (3), this latter process would result into a stochastic source term for the excitations of the fluid[50]. Experimentally, we find that the relative intensity of the excitations is very small as compared with the coherent field, so that a linearized treatment around the steady state is an accurate description.

This model describes a condensate on top of which excitations of frequencies $\omega_e(k)/2\pi$ are present, that can be classified into two distinct regimes. On one hand, the excitations of zero frequency $\omega_e = 0$ correspond to the formation of a static spatial

pattern as generated, e.g., by a condensate flowing against an obstacle. It is governed by the total population of the condensate plus reservoir, i.e., the total blueshift $\hbar\omega_{BS} = \hbar g|\psi|^2 + \hbar g_R n_R$. In this limit, our model can be mapped onto a GGPE, in which an effective interaction term defined as $g_{eff,s} = g + g_R\gamma_{in}/\gamma_R$ such that $g_{eff,s}n = \hbar\omega_{BS}$ is used, and contains the reservoir contribution. Then, the known properties of the GGPE in the static limit[13] do apply. In particular, the suppression of polariton scattering due to interaction with an obstacle is expected to happen below a critical velocity $v_c = \sqrt{\hbar\omega_{BS}/m}$, which is in agreement with the pioneering experimental reports of superfluidity[20,21]. On the other hand, in the regime where excitations have a frequency $\omega_e/2\pi \gg \gamma_R$, as is the case in this work, the reservoir response is too slow to follow the condensate fluctuations. Our model can still be mapped onto a GGPE but with a different effective interaction $g_{eff,d} = g$, and $g_{eff,d}n < \hbar\omega_{BS}$. In this nonzero frequency limit, the speed of sound $c_s$ of the gapless sonic state is well-defined and is governed solely by the polariton condensate fraction: $c_s \simeq \sqrt{\hbar gn/m}$. Note that for this particular sonic state, $c_s < v_c$, which means that the standard prediction of Landau's critical velocity must be modified. This is due to the fact that excitons in the reservoir are fixed in the material frame, which in turn affects the Galilean relativity features of the coupled system. Based on these arguments, the contributions to the blueshift due to polariton–polariton and polariton–reservoir interactions can be separated by comparing the polariton interaction extracted from the measured speed of sound $\hbar gn = mc_s^2$ and the total blueshift $\hbar\omega_{BS}$.

In order to simulate the experimental measurements, we used the vectorial version of the model, and determined the energy-momentum power spectrum of the condensate wavefunction by convolving the Bogoliubov matrix response function with a stochastic Gaussian noise simulating the coupling with the phonon bath. We then extracted the DR by fitting these data using the same method as for the experimental data. We obtained a quantitative agreement with the experimental DR both in WPA (thick red/magenta line in Fig. 3b/d) and WPB (thick red/magenta line in Fig. 3g/i). The only fitting parameter is the steady-state ratio of condensate to reservoir interaction energy. Note that in this vectorial theory, the DRs exhibits a total of five branches: two on the positive energy side, two on the negative side, and a flat one that we do not have access to in this experiment (cf. Supplementary Note 8). As a result, the calculated DR that we use to fit our measurements consists of a nontrivially weighted contribution of two branches originating from the co- and crossed-polarized condensate, which also contribute to the linear part of the dispersion.

The best fit is obtained for $\rho = n/(n_R + n) = 46\%[38\%; 53\%]$ for WPA, and $73\%[60\%; 85\%]$ for WPB, where we used $g_R = g/|X|^2$, where $|X|^2 = 0.58$ and $|X|^2 = 0.38$ for WPA and WPB, respectively, $n = |\psi_x|^2 + |\psi_y|^2$ and the indices $x, y$ refer to the co- and cross-polarized components, respectively, and the bracket is the 95%-confidence interval. The line thickness of the calculated DRs in Fig. 3b, d (WPA) and in Fig. 3g, i (WPB) show the area overlapped by the theoretical DRs within this interval. In the limit of vanishing birefringence, $|\psi_y|^2 \ll |\psi_x|^2$, the characteristic speed of sounds amount to $c_x/v_c = 0.42 \pm 0.05$ for WPA and $c_x/v_c = 0.56 \pm 0.12$ for WPB (see details in Supplementary Note 6). In addition, we can use this knowledge of the condensate contribution to the total blueshift to estimate the polaritons–polariton interaction constant, excluding interaction with the reservoir. Upon normalizing by the number of quantum well, we thus find $\hbar g_T = 8 \pm 2\ \mu eV\cdot\mu m^2$ in WPA, which is mostly

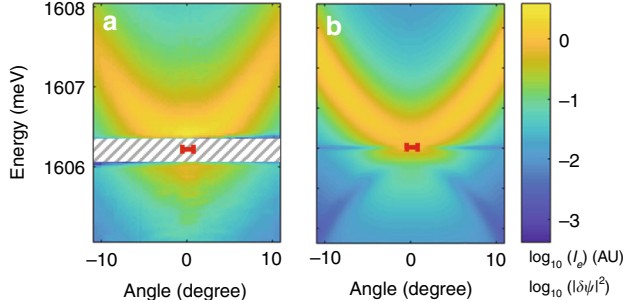

**Fig. 4** Superfluid excitations density analysis. **a** Measured cross-polarized EPL measurements in the upper branch of $I_0(P)$ close to the switch-down point, with an enhanced signal-to-noise ratio of 300. **b** Calculated angle- and energy-resolved cross-polarized EPL from the full vectorial theory assuming a condensate fraction of $\rho = 50\%$ and experimental parameters from WPA. The intensity is color coded on a four decades logarithmic scale, and the laser energy and angular spread is shown as a red segment. The hatched rectangle shows the spectral range rejected by the notch filter

in-line with the values previously reported in the literature for microcavities of similar or closely related design[51–53]. Note that this value is to be considered as an order of magnitude as the experiment was not optimized for this quantitative derivation of $g_T$ (see details in Supplementary Note 7).

Remarkably, our model also recovers the fact that in our measurements, the ghost branch emission is strongly suppressed (cf e.g., Fig. 2d, e). We double-checked this feature in the measurement shown in Fig. 4a in which the signal-to-noise was enhanced to 300, and in which emission from the ghost branch is not found either. Our model explains the two different origins contributing to this suppression: (i) the ghost branch contribution vanishes when the incoherent (reservoir) fraction of the fluid increases, an effect which is common to the scalar situation. (ii) In cross-polarized detection geometry, the different polarization contributions to the excitations interfere in a nontrivial way as is visible in Eq. (16) in Supplementary Note 8. This interference can either enhance or, as in our case, suppress the ghost branch emission. Note that this mechanism is not directly related to the one discussed in ref. [54], which refers to a different pumping scheme. Note also that in ref. [32], special care was paid to reduce the excitonic disorder (i.e., the reservoir contribution)[49,55], such that the emission from the ghost branch could be detected, albeit in a regime dominated by the free-particle properties.

In summary, we obtained a direct measurement of the dispersion relation of the collective excitations on top of a coherently driven polariton condensate. This measurement can be directly used as the key experimental ingredient required to test the system against most superfluidity criteria. Our analysis also provides a deep insight into the unique two-components nature of this quantum fluid and its elementary excitations: partly polariton condensate and partly incoherent reservoir. In particular, we have checked that the speed of sound, which is a key characteristic of collective excitations in quantum fluids, is strongly reduced by the presence of the reservoir. More generally, we have established theoretically and experimentally that the reservoir involvement results in two different regimes of collective excitations. On one hand, the spatial density patterns resulting from static collective excitations like in historical frictionless flow experiments[20,21], involve the total blueshift due to polaritons and reservoir. On the other hand, finite-frequency collective excitations as investigated here, are governed only by interactions between condensate polaritons.

We are currently investigating the extent and magnitude of these consequences from both the theoretical and the experimental point of view. For instance, the hydrodynamic nucleation of vortices studied in refs. [24,56–58] is a time-dependent phenomenon, for which one can naturally expect strong modifications due to the coupling of the coherent polariton fluid to the much slower degrees of freedom of the reservoir that strongly affects its Galilean relativity properties. From a quantum-optical perspective, the additional blueshift of the polariton state due to the incoherent reservoir may be playing a role in the experimental recent two-body correlation experiments[51,59]. Along the same lines, the presence of an incoherent excitonic reservoir reservoir is a convincing argument[60] to explain the overestimated polariton–polariton interaction constant in ref. [61]. In this context, our spectroscopic measurement of the collective excitations provides an accurate way to extract the polariton–polariton interactions contribution to the total interaction energy, and isolate it from the reservoir contribution.

## Methods

**Detail of the microcavity.** The GaAs/AlAs microcavity used in this experiment is identical to that used in ref. [62]. It features a quality factor $Q = 3000$. This relatively low quality was chosen on purpose as it satisfies two conflicting requirements: (i) the need to have a collective excitation energy window that largely exceeds the instrumental resolution of 70 μeV, and (ii) the need to keep the laser intensity low, which requires the linewidth to be not too much smaller than $\Delta$. The heavy-hole and light-hole excitonic transitions energy are at $E_{hh} = 1612.05$ meV and $E_{lh} = 1644$ meV at $T = 30$ K, and the corresponding Rabi splittings resulting from the strong coupling regime with the cavity mode of energy $E_{c0}$ are $\hbar\Omega_{hh} = 15$ meV and $\hbar\Omega_{lh} = 12.5$ meV, respectively. The microcavity is intentionally wedged in order to tune $E_{c0}$ with respect to $E_{hh}$. The background index of the microcavity is $n_{bg} = 3.65$. The microcavity is placed in the vacuum chamber of a temperature-tunable Helium flux cryostat. The temperature is set at $T = 30$ K, found as an optimum of thermal phonons population and polariton linewidth. WPA (WPB) is characterized by a detuning $\delta = E_{c0} - E_{hh} = +1.25$ meV ($\delta = -1.82$ meV). The lower polariton mode exhibits a full-width at half-maximum of $\hbar\gamma_c = 0.4$ meV, and a reservoir decay energy $\hbar\gamma_R = 1.6$ μeV in both WPs.

**Parameters of the resonant laser excitation.** The polariton fluid is driven resonantly with a single longitudinal mode CW Ti-Sapphire laser of 5 MHz linewidth, and linearly polarized with a high purity. The laser beam is shaped with pinholes into a spatially Fourier-limited Gaussian mode of 50 μm diameter as measured on the surface of the microcavity, and a corresponding $\delta\theta = \pm 2°$ angular spread (i.e., $\delta k_\parallel = \pm 0.28$ μm$^{-1}$) in momentum space. WPA is excited at normal incidence, while WPB is excited with a $-1°$ incidence angle. The detuning $\Delta$ between the polariton mode and the CW laser are $\Delta = 0.79$ meV (WPA), and $\Delta = 0.47$ meV (WPB).

**Full vectorial theory.** Owing to its polarization degree of freedom, the condensate field is a vectorial field, characterized by its two components $\psi_\sigma(\mathbf{r})$ in the $\sigma = x, y$. Each of these component is described by a GGPE equation coupled with the other, and additionally coupled to the dark-exciton reservoir via its density $n_R$:

$$i\partial_t \psi_x = \left[\omega_{LP}(\hat{\mathbf{k}}) - \frac{\alpha}{2}\cos(2\Theta) + \frac{g_T + g_S}{2}|\psi_x|^2 + g_T|\psi_y|^2 + g_R n_R - i\frac{\gamma_c + \gamma_{in}}{2}\right]\psi_x$$
$$- \frac{\alpha}{2}\sin(2\Theta)\,\psi_y - \frac{g_T - g_S}{2}\psi_x^*\psi_y^2 + F$$

(5)

$$i\partial_t \psi_y = \left[\omega_{LP}(\hat{\mathbf{k}}) + \frac{\alpha}{2}\cos(2\Theta) + \frac{g_T + g_S}{2}|\psi_y|^2 + g_T|\psi_x|^2 + g_R n_R - i\frac{\gamma_c + \gamma_{in}}{2}\right]\psi_y$$
$$- \frac{\alpha}{2}\sin(2\Theta)\,\psi_x - \frac{g_T - g_S}{2}\psi_y^*\psi_x^2$$

(6)

$$\partial_t n_R = -\gamma_R n_R + \gamma_{in}(|\psi_x|^2 + |\psi_y|^2),$$

(7)

where $\omega_{LP}(\hat{\mathbf{k}}) = \omega_{LP,0} - \frac{\hbar}{2m}\nabla^2$ and $\hat{\mathbf{k}} = -i\nabla$, with $m$ the effective polariton mass, $\alpha \sim 0.1 \pm 0.05$ meV is the birefringence splitting measured at normal incidence, and $\Theta \simeq 19°$ is the angle between the birefringence axes and the cross-polarized measurement basis $x, y$. $g_T$ and $g_S$ are the triplet and singlet coupling constants respectively, and we take $g_S = -0.1\, g_T$ with $g_T > 0$. The measurement of the steady-state dispersion relation provides us with a measurement of the condensate interaction energy $\bar{g}(|\psi_x|^2 + |\psi_y|^2)$, and on the reservoir interaction energy $g_R n_R = g_R(|\psi_x|^2 + |\psi_y|^2)\gamma_{in}/\gamma_R$. An independent knowledge of $\bar{g}$, $g_R$ or $(|\psi_x|^2 + |\psi_y|^2)$ requires additional assumptions or measurements, like the exact impinging drive power at the hysteresis switch up point. The pump field is $F(\mathbf{r}, t) = F_0 e^{i\mathbf{k}_p \cdot \mathbf{r} - i\omega_p t}$. The other parameters are the reservoir filling rate $\gamma_{in}$ from the condensate, the reservoir decay rate $\gamma_R$ and the condensate radiative loss rate $\gamma_c$. The properties derived from this theory are discussed in details in Supplementary Note 8.

**Determination of the experimental dispersion relation and its confidence interval.** The experimental dispersion relations are extracted from the measurements of $I_e(k, \omega)$: For each column $i$ of $I_e(k_i, \omega_j)$ that contains the spectrum at wavevector $k_i$, the EPL emission peak is fitted with a Lorentzian peak. From this fit and its goodness, we get the central frequency $\omega_{0,i}/2\pi$ of the peak, and its 95% confidence interval $\delta\omega_i$, respectively. The thus obtained ensemble $\omega_{0,i}(k_i)$ is the extracted dispersion relation, and $|\delta\omega_i(k_i)|_i$ is the 95% confidence interval computed from the optimization algorithm used in the fitting procedure. This procedure is described step by step for WPA point "3" in Supplementary Note 2.

## Data availability

The data that support the findings of this study are available from the corresponding author upon reasonable request.

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

## Acknowledgements

The authors wish to thank M. Wouters for crucial discussions in the early stage of this project. P.S, J.B., A.A.G., and M.R. acknowledge funding from the french ANR contract "QFL" (ANR-16-CE30-0021). I.C., J.B., A.L., and A.A. acknowledge funding from the H2020-FETFLAG-2018-2020 project "PhoQuS", (nb 820392). A.M. acknowledges funding from the french ANR contract "SuperRing" (ANR-15-CE30-0012). J.G.R. acknowledges the "ETIUDA" program from the polish NCN.

## Author contributions

P.S. and J.-G.R. carried out the measurements, I.A., I.C., and A.M. developed the theoretical interpretation and modelization. M.R. and A.M. led the project. J.B., A.L., and A.A. provided the sample and participated with all the authors to the the discussion of the results, the data analysis, and the redaction.

## Additional information

**Competing interests:** The authors declare no competing interests.

