## [Peer Review File · Nature Communications]

Reviewers' comments:

Reviewer #1 (Remarks to the Author):

This manuscript presents an investigation of dispersion relation for thermal excitations of a resonantly pumped exciton-polariton condensate hosted by a semiconductor microcavity. By comparing the measured dispersion with that predicted by theory in two limiting cases (non-interacting, single particle regime and interacting condensate regime), the authors make a case for the presence of an incoherent excitonic reservoir which is populated by scattering of exciton polaritons with phonons. Although previous studies in the field (referenced in this paper) have hinted at the possibility of reservoir excitation even under the resonant pumping condition, this new study presents the most compelling and clear evidence that this is indeed the case.

I find the investigation presented in the paper to be potentially impactful (in the area of exciton-polariton physics), and very thorough. However, several points need to be clarified before I can make a recommendation regarding the publication of this work in Nature Communications.

1. The authors calling the regime of condensation that they investigate a "high-density regime". Yet, there is no indication that this is the case, nor any indication of how strong is the pump relative to the condensation threshold. A plot of PL intensity as a function of pump power at the location of the condensate (in momentum space) would be useful in this regard.
2. As far as I understand, filtering out the emission from the condensate is critical to observing the emission due to elementary excitations. For that matter, I do not understand the placement of the filter in Fig. 2(d).
3. The authors interpret Fig. 2(a) as a closed-up hysteresis loop. However, the scalar theoretical model (3,4) does not admit bistable behaviour, which makes the narrative somewhat confusing. What is the physical reason for the bistability of this system when pumped at a normal incidence (as opposed to an OPO configuration)?
4. Somewhat related to p. 3: How are the I(P) curves in Fig 2(a) affected by the choice of the working point (WP) on the sample?
5. As stated in the manuscript, the two WPs chosen by the authors allow them to investigate the dispersion of excitations at different strength of interactions (g). Presumably, this is due to a cavity wedge in the sample, which leads to a shift of the cavity photon relative to the exciton resonance? If so, this point needs to be explicitly stated.
6. Further to p.5, WPB should correspond to an exciton-polariton which is more photonic (weaker interacting). This seems to be supported by the difference in the effective mass of the exciton polariton apparent in Fig. 3. However, the value of the blueshift at WPB is comparable with that at WPA (see Fig. 3). Can authors comment on the reason for this? Also, there should be a difference in the radiative decay rates at the two points (hence the linewidth) affecting the experimentally extracted dispersions and the corresponding error interval.
7. In the theoretical modelling sections, the authors assume that the polariton-reservoir interaction constant is twice the polariton-polariton interaction constant. While this assumption is justified for 50% excitonic fraction (as determined by the Hopfield coefficients), this cannot be the case for both WPA and WPB.
8. Suppression of the ghost branch PL in the nonequilibrium condensate has been recently discussed in Hanai et al Phys. Rev. B 97, 245302 (2018). Can the authors comment on the relevance of this theory to their results?

9. In conclusions, the authors claim that their measurement allows one to isolate the blueshift caused by the reservoir from that caused by polariton-polariton interactions. In my opinion, this statement is not clearly justified since the whole study argues that both interaction effects (polariton-polariton interactions and polariton-reservoir interactions) contribute to the shape of the excitation branches.

Additional technical comments:

p. 1: "The corresponding extracted... is labelled '0' in Fig. 3 a." This should read "Fig. 2a".

p. 4: "Note that under non-resonance..." Should be "non-resonant".

p. 5: "...condensate excitations have changed from free-particle to collective as a result of strong interactions" – I would suggest to re-phrase this. The strength of interactions doesn't change, density does.

p.5: sentence after Eq. (1): "...is known from '0'" – Do the authors mean "from the measurement at point '0'"?

p. 5: The notation convention for superscripts (e.g. "(A_i)") needs to be specified.

p. 6: "In this limit, our model can be mapped onto the GGPE upon identifying..." This statement is unclear. Isn't the whole point here that the blueshift cannot be attributed to the condensate density alone?

p. 6: Two different notations "n" and "n_c" seem to be used for the same quantity (condensate density).

SI: Ref. 6 is duplicated in Ref. 11

Reviewer #2 (Remarks to the Author):

In this manuscript, the authors claim to measure the excitation spectrum of polariton condensates and they compare it with a very accurate theoretical model. The focus of the manuscript is the demonstration of the presence of a reservoir made of "dark" excitons with much longer lifetimes, as compared to the bright excitons that constitute the condensate. Such a reservoir affects the energy of the condensate having significant repercussion on the spectrum of excitations in these driven/dissipative quantum fluids. The work is very timing and the interest goes beyond a mere difference on the superfluid characteristics of a polariton quantum fluid since the presence of such a reservoir would also respond to many debates on the nature of polariton interactions, at present a topic under the spotlight. The presence of such a reservoir is evidenced by looking at the speed of sound. Interactions with the reservoir cannot contribute to the sound speed, which is consequentially smaller than what estimated by measuring the blueshift, as done in previous literature. Therefore, this work suggests a way out by separating the slow excitonic contribution, even of optically inactive excitons, from the polariton-polariton repulsion. The interest of these findings is twofold. First, these results address the role of a long-lived reservoir on the hydrodynamics of quantum fluids, which attracts a broad interest also beyond the polariton community. Second, only polariton-polariton interactions are useful from a quantum optics perspective and the debate about the actual interaction strength is a very timely topic. For these reasons the work deserves to be considered for publication in Nature Communications. However, before a final decision can be made, the following points needs to be addressed.

Major points:

- An important point is to assess if the differences between the calculated and measured excitation spectra at high k can solely be attributed to the presence of "dark" excitons. One of the major doubts could come by the non-uniform emission below the laser spot. First of all, from Fig. 6S it seems that the laser gaussian is not much larger than the region under the circular mask. This could lead to strong inhomogeneities. The UB in Fig.6S, for instance, shows quite noticeable inhomogeneities in space within the circular mask used in the detection path, showing darker and brighter regions. Does the numerical modeling see a similar inhomogeneity?
- The measurement in the Fourier plane averages over different spatial regions with possibly different regimes even within the circular mask. The signal coming from brighter regions may be affecting only the part of the excitation spectrum close to $k=k_p$ (basically at $k=0$), while darker regions, with lower interactions, contribute substantially to the dispersion at higher k .
- Note that the linear dispersion is shown for angles $<5^\circ$, while it is almost parabolic for angles $>5^\circ$. At the same time, it is particularly difficult to estimate the exact dispersion for angles $<5^\circ$ from the condensate due to the energy filtering. As shown in Fig. S2, for angles $<4.7^\circ$ the peak of the Lorentzian is masked, and we have to rely only on the Lorentzian fitting. If multiple resonances are present below 5° , it would be difficult to distinguish between them.
- The spatial filtering over smaller regions will probably limit irremediably the resolution in the Fourier plane, but still a discussion about the way one can exclude this possibility seems required.
- The interpretation of the results is slightly complicated by the cross-polarized experiments. The birefringence is useful to filter out the scattered laser but a more sophisticated vectorial model is needed. The presence of multiple branches and the partial optical access makes less clear which branches are populated. This should be explained more extendedly and probably a plot of the multiple branches could help to follow the description.

Minor points:

- Is the healing length also reduced? would it be possible to measure the healing length and compare it with the speed of sound?
- Which other systems may show the dynamic separation of the reservoir from the excitations in the superfluid phase?
- Is the dispersion of excitations at high- k blue-shifted from the parabolic dispersion by the total blueshift, only by the polariton part or a combination of the two?
- How is the energy of the condensate precisely estimated if it is not the same as the central laser frequency?
- How the frequency of the excitations is defined in pg.6: "In the regime where excitations of frequencies.."? Is this giving a cutoff wavevector for the crossover from the rigid-blueshift to the decoupled-reservoir-condensate regimes? How much is it? How does it change by changing γ_r ?
- The small Q factor and spatial inhomogeneity are a limiting factor for the maximum precision achievable even in this accurate experimental realization. Can longer lifetime be useful in the measurements of the actual speed of sound?
- The detuning used in Fig2a should be indicated in the caption.
- At pg.4, Fig.2c should be changed with Fig.2a in the sentence: "two states on the curve $I_0(P)$ labeled '1' and '2' respectively in Fig.2.c"
- Does this sentence: "Finally, the speed of sound that would be observed at the sonic point can be deduced from ρ , .., $c/v=0.5$ and $c/v=0.6$ " refer to WPB at power 2 and 3?
- Pg.3 of SI: "and the can lie"
- Pg.9 SI: corresponding TO the particle and hole branch

Reviewer #3 (Remarks to the Author):

The manuscript "Dispersion relation of the collective excitations in a resonantly driven polariton

fluid" by Stepanov et al. combines theoretical and experimental efforts to study the dispersion relation of the collective excitations on top of a coherently driven polariton condensate. The understanding of the spectrum of such excitations is fundamentally important for our understanding of the physics of exciton-polaritons.

However, before I can recommend the paper for publication I would ask the authors to clarify several points about the theory presented.

1. γ_{bk} in Eq (4) describes the rate with which reservoir excitons are converted back into polaritons. Should not in this case γ_{bk} be multiplied by $|\psi|^2$?

Apart from this, the justification for dropping γ_{bk} from equations is that it is much smaller than γ_c . However, both γ_R and γ_{bk} are smaller than γ_c , so γ_{bk} should be compared to γ_R not γ_c when deciding the relevance of various terms. Furthermore, if γ_{bk} is now multiplied by the condensate density it may no longer be negligible for sufficiently large densities. It would help to provide the analysis of what ranges of γ_{bk} are expected.

2. One of the conclusions of the manuscript is that "the spectroscopic measurement of the collective excitations provides a new and accurate way to extract the polariton-polariton interactions contribution to the total interaction energy, and isolate it from the reservoir contribution." Indeed, it seems that the value of g should immediately follow from the analysis presented. Can the authors please provide the estimates on the polariton-polariton interaction strength that follows from their analysis and relate or contrast their findings with known estimates? In particular, what is the value of g_T used in Section C and Supp. Inf VII?

3. Have the same values of γ_R , γ_c , γ_{in} been used in all simulations? It would be good to have their values stated and discussed in the main part of the paper. Supp. Inf. gives these as $\gamma_R = \gamma_{in} = 0.015 \text{ meV}$, $\gamma_c = 0.4 \text{ meV}$. However, the experimental decay rates give the ration of about 8 between the life-time of polaritons and reservoir excitons (58ps and 400 ps). Where does the discrepancy come from?

4. The main advantage of the presented theory is that it explains factor 2 discrepancy in the measured speed of sound, but to claim quantitative correctness of the model the agreements in other measured characteristics should be shown.

REFEREE 1

We thank the referee for his/her positive assessment of our work, by finding it potentially impactful in the area of polariton physics, and for his/her thorough review of our work. We give below a detailed answer to each comments as well as the list of corrections implemented in the manuscript and in the Supplementary information.

- 1- *The authors calling the regime of condensation that they investigate a “high-density regime”. Yet, there is no indication that this is the case, nor any indication of how strong is the pump relative to the condensation threshold. A plot of PL intensity as a function of pump power at the location of the condensate (in momentum space) would be useful in this regard.*

In this resonant excitation configuration, unlike in the non-resonant excitations case, and unlike in the OPO configuration, we have only one densely populated polariton population at all time: the one driven by, and resonant with the laser. The laser thus fixes its energy, coherence, density, and width in momentum space (tuned at 2° HWHM as expressed in angular spread), and we refer to it as the “condensate”.

This resonant pump configuration is that used for instance in [45,46] and in the superfluidity measurements like in [14]. Thus, there is no condensate formation, or condensation with a well-defined threshold, in the sense of the spontaneous symmetry breaking as it occurs under non-resonant excitations or as for the signal state in the OPO configuration.

The expression “high-density regime” refers to the states of the polariton fluid in which the interaction energy exceeds the linewidth. This is the case of the states labelled 2 and 3 in Fig.2 (WPA), Fig.3 (WPA and WPB), and Fig.S4 (WPC). As a property of the hysteretic regime, in the upper branch of $I(P)$, the blueshift equals or exceeds the laser detuning with respect to the addressed polaritons state $\Delta = \omega_l - \omega_0$ (cf. right axis of the figure below). In our experiments $\Delta = 0.79\text{meV}$ (WPA), $\Delta = 0.47\text{meV}$ (WPB), $\Delta = 1.04\text{meV}$ (WPC). The blueshifts investigated in WPA thus are $\hbar\omega_{BS} = 0.85\text{meV}$ (state 2) and $\hbar\omega_{BS} = 0.90\text{meV}$ (state 3). In WPB, $\hbar\omega_{BS} = 0.55\text{meV}$ (state 2) and $\hbar\omega_{BS} = 0.61\text{meV}$ (state 3). In WPC, $\hbar\omega_{BS} = 1.11\text{meV}$ (state 2) and $\hbar\omega_{BS} = 1.14\text{meV}$ (state 3). For comparison, the linewidth amounts to $\hbar\gamma = 0.4\text{meV}$. This “high density regime” is that for which the dispersion relation exhibits a measurable shape change, due the crossover from one-body to collective excitations.

- ➔ **In the main text, we now give an explicit definition of what the word “condensate” and “high-density regime” refers to.**

2. *As far as I understand, filtering out the emission from the condensate is critical to observing the emission due to elementary excitations. For that matter, I do not understand the placement of the filter in Fig. 2(d).*

The spectral filter is tuned at the condensate energy, which is always equal to that of the laser. This is the energy where the light undergoes strong coherent scattering that needs to be blocked from the detector.

Note that under resonant drive, as predicted in early theoretical works (e.g. refs [11,13]), the dispersion relation of the elementary excitation does not necessarily pass by the laser frequency: it can be gapped with respect to it.

3. *The authors interpret Fig. 2(a) as a closed-up hysteresis loop. However, the scalar theoretical model (3,4) does not admit bistable behaviour, which makes the narrative somewhat confusing. What is the physical reason for the bistability of this system when pumped at a normal incidence (as opposed to an OPO configuration)?*

Fig. R1 (new Fig.S11 in the SI): Calculated polariton density n versus pump intensity $|F|^2$ for three different values of the laser detuning Δ . When $\Delta > \hbar\gamma\sqrt{3}/2$ (red plot), $n(|F|^2)$ exhibits a bistable behavior.

We kindly disagree with the referee. Eq.(3,4) do exhibit a bistable behavior when the laser drive frequency in the drive term F is blue-detuned from the addressed polariton resonance by $\Delta \geq \hbar\gamma\sqrt{3}/2$. Even though the equations do indeed resemble very much the non-resonant excitation theory, there is one key difference: the position of the pump term “ F ”. In our case indeed, F enters Eq.(3): it is a coherent pump term for the polariton field, while in the non-resonant case it enters Eq.(4) and constitute an incoherent pump term that generates a population of excitons in the reservoir. This seemingly minor difference is responsible for the appearance of bistability in our model, whose bistability features resemble quite closely the ones of the standard, reservoirless, coherent driving case discussed e.g. in Sec.IV-B of Ref [11].

- ➔ **We have added a more detailed description of this model in the SI (VIII.A last subsection, and new Fig.S11) showing in particular the bistable regime.** An example is shown in figure R1 above, where the following parameters have been used: $g_T=10\mu\text{eV}\cdot\mu\text{m}^{-2}$; $g_R=1.7g_T$; $\hbar\gamma=0.4\text{meV}$; $\hbar\gamma_R=1.6\mu\text{eV}$; $\gamma_R=1.8\gamma_{in}$. Δ is the laser detuning with respect to the addressed polariton state. n is the steady-state total polariton population, $4|F|^2/\gamma^2$ is the normalized pumping rate. The hysteretic regime is achieved when $\Delta \geq \hbar\gamma\sqrt{3}/2$.

4. *Somewhat related to p. 3: How are the $I(P)$ curves in Fig 2(a) affected by the choice of the working point (WP) on the sample?*

In the experiment, the $I(P)$ curves indeed vary a bit when moving from one WP to another, due to the small change in interaction strength on one hand, and to the spatial fluctuations in the plane of the microcavity (as discussed in detail in the reply to referee 2 pt 1 below). The result is a moderate difference in the jump-up pump power in the $I(P)$ curves, as can be seen for instance by comparing Fig.2.a of the main text (point WPA), and Fig.S5.a in the SI (a point similar to WPA in detuning, but at another position on the sample, i.e. seeing another local disorder). Note that unlike the horizontal axis (excitation power), the vertical axis (the transmitted light intensity) is not directly comparable from point to point due to slightly different detection conditions.

5. *As stated in the manuscript, the two WPs chosen by the authors allow them to investigate the dispersion of excitations at different strength of interactions (g). Presumably, this is due to a cavity wedge in the sample, which leads to a shift of the cavity photon relative to the exciton resonance? If so, this point needs to be explicitly stated.*

The text was possibly not clear enough: the two working points WPA and WPB correspond to two different situations in which the condensate is excited with a zero and nonzero pump wavevector, respectively. The referee is however correct that the two working points have a slightly different detuning, which indeed means a slightly different interaction strength.

As suggested by the referee, we now mention the true (experimentally determined) excitonic fraction of each WPs in the main text, and we have recalculated accordingly the values of the condensate fractions and speed of sounds in each WPs, leading to corrections of the order of 10%.

➔ **These corrected values are now given in the main text, as well as in the SI section VI, and the corresponding Fig.S9.**

6. *Further to p.5, WPB should correspond to an exciton-polariton which is more photonic (weaker interacting). This seems to be supported by the difference in the effective mass of the exciton polariton apparent in Fig. 3. However, the value of the blueshift at WPB is comparable with that at WPA (see Fig. 3). Can authors comment on the reason for this?*

In WPA, $\Delta = 0.79$ meV, $\hbar\omega_{BS} = 0.85$ meV and $\hbar\omega_{BS} = 0.90$ meV, in states 2 and 3 respectively. As WPB is indeed more photonic, we have chosen a smaller Δ ($\Delta = 0.47$ meV) in order to deal with smaller blueshifts: $\hbar\omega_{BS} = 0.55$ meV and $\hbar\omega_{BS} = 0.61$ meV in states 2 and 3 respectively. This configuration allows us to deal with a similar polariton density in WPA and WPB.

7. *...Also, there should be a difference in the radiative decay rates at the two points (hence the linewidth) affecting the experimentally extracted dispersions and the corresponding error interval.*

Linewidth measurements in WPA and WPB can be found in Fig.S6 of the SI (black line for the non-interacting regime). Since the difference in detuning δ between WPA and WPB is small as compared to the Rabi splitting (cf. point 5 above), the linewidth difference is equally small (black line in Fig.S6.a and Fig.S6.b). Our measurement show that the linewidth $\hbar\gamma$ in WPA and WPB differ only by ~ 0.03 meV.

8. *In the theoretical modelling sections, the authors assume that the polariton-reservoir interaction constant is twice the polariton-polariton interaction constant. While this assumption is justified for 50% excitonic fraction (as determined by the Hopfield coefficients), this cannot be the case for both WPA and WPB.*

This is correct: cf. Point 5 above. The sentence “We assumed that $g_R = 2g$ due to polariton 50% excitonic fraction, such that...” was indeed misleading in this regard **and has been removed**. As indicated in the answer to point 5 above, we have improved the theoretical modeling to include the dependence of the interaction strength on the excitonic fraction into account.

Suppression of the ghost branch PL in the nonequilibrium condensate has been recently discussed in Hanai et al Phys. Rev. B 97, 245302 (2018). Can the authors comment on the relevance of this theory to their results?

The theoretical work mentioned by the referee discusses the excitation spectrum of a polariton condensate under non-resonant excitation. Like in the resonant case, the excitations of the condensate are of collective nature and thus exhibit a ghost branch. However, the nature and dispersion relation of these excitations are different: at long wavelength and under non-resonant drive, the excitations consist in a diffusive Goldstone mode, while under resonant drive, the excitations are essentially sonic or gapped. Even though the relation to our work is not a direct one, **citation of this interesting work has been added to the revised text.**

9. *In conclusions, the authors claim that their measurement allows one to isolate the blueshift caused by the reservoir from that caused by polariton-polariton interactions. In my opinion, this statement is not clearly justified since the whole study argues that both interaction effects (polariton-polariton interactions and polariton-reservoir interactions) contribute to the shape of the excitation branches.*

Indeed, the model includes both interaction effects. However, the calculation shows that, due to the different nature of the two fluids, at non-zero frequency only the polariton condensate component contributes to the shape of the dispersion relation. This is a central point of our work.

- ➔ **In the revised version, we have added a sentence stating more clearly how the two components of the blueshift can be separated using the experimental data.**

Editorial comments

p. 1: “The corresponding extracted.... is labelled ‘0’ in Fig. 3 a.” This should read “Fig. 2a”.
Fig.3.a is in fact the correct reference (cf. the green dispersion relation in Fig.3.a)

p. 4: “Note that under non-resonance...” Should be “non-resonant”.
Corrected

p. 5: “..condensate excitations have changed from free-particle to collective as a result of strong interactions” – I would suggest to re-phrase this. The strength of interactions doesn’t change, density does.
Sentence changed into “from free-particles to collective as a result of strong interaction energy”

p.5: sentence after Eq. (1): “..is known from ‘0’” – Do the authors mean “from the measurement at point ‘0’”?
Sentence changed into “is known from the DR measurement at point ‘0’ ”

p. 5: The notation convention for superscripts (e.g. “(Ai)”) needs to be specified.
The notations have been homogenized. The notation Δ_{BS}^{Ai} was superfluous and has been replaced with $\hbar\omega_{BS}$ as everywhere else.

p. 6: “In this limit, our model can be mapped onto the GGPE upon identifying...” This statement is unclear. Isn’t the whole point here that the blueshift cannot be attributed to the condensate density alone?

Our model accounts explicitly for the reservoir contribution and allows describing any states of the fluid, and any type of excitations on top of the condensate. In the discussion pointed out by the referee, we show that in two limiting situations, this model can be mapped onto two different GGPEs, featuring two different effective interaction strengths $g_{\text{eff},s}$ and $g_{\text{eff},d}$:

- In the limit of static excitations (i.e. when the excitations frequency $\omega_e/2\pi = 0$), our model can be mapped onto an equation that has the form of a GGPE, and in which the contribution of the reservoir is hidden into an effective interaction constant $g_{\text{eff},s} = g + g_R\gamma_{\text{in}}/\gamma_R$ that replaces GGPE's g .
- This mapping then breaks down when $0 < \omega_e < \gamma_R$, a situation that can be described only with the full model.
- Then, it takes again the form of a GGPE when $\omega_e > \gamma_R$, but with a different effective interaction strength: $g_{\text{eff},d} = g \neq g_{\text{eff},s}$.

As discussed in the main text, the interest of discussing these limiting cases is that (i) they are often relevant in past work from the literature (in particular the static limit), and (ii) we can use the generic properties of the GGPE previously derived in the literature, to understand the fluid behavior in the regime where the mapping stands. Note however that only our model successfully describes consistently the whole excitation spectrum irrespective of the frequency range.

➔ **This paragraph has been extended to present this discussion more clearly.**

p. 6: Two different notations “n” and “n_c” seem to be used for the same quantity (condensate density).

Corrected

SI: Ref. 6 is duplicated in Ref. 11

Corrected

REFEREE 2

We thank the referee for his/her very positive assessment of our work, in finding that it “deserves to be considered for publication in nature communication”, and for drawing our attention on important points in our work. We give below a detailed answer to each comments as well as the applied corrections.

1. *An important point is to assess if the differences between the calculated and measured excitation spectra at high k can solely be attributed to the presence of “dark” excitons. One of the major doubts could come by the non-uniform emission below the laser spot. First of all, from Fig. 6S it seems that the laser gaussian is not much larger than the region under the circular mask. This could lead to strong inhomogeneities. The UB in Fig.6S, for instance, shows quite noticeable inhomogeneities in space within the circular mask used in the detection path, showing darker and brighter regions. Does the numerical modeling see a similar inhomogeneity?*

The measurement in the Fourier plane averages over different spatial regions with possibly different regimes even within the circular mask. The signal coming from brighter regions may be affecting only the part of the excitation spectrum close to $k=k_p$ (basically at $k=0$), while darker regions, with lower interactions, contribute substantially to the dispersion at higher k .

We agree with the referee that this is an important question. The excitations spot indeed has a Gaussian profile, and Fig.S5.b-d suggest the presence of density fluctuations within the region selected by the circular mask. These fluctuations are consistent with the presence of a weak disordered potential as is shown in Fig.2.b in the main text.

Let us first emphasize an important fact: Fig.S5.b-d plots only the cross-polarized condensate density distribution in real space $n_y(r)$, which is not the total density $n_x(r) + n_y(r)$ that governs the interactions and the shape of the dispersion relation. As a result, while Fig.S5.c,d correctly suggests the presence of spatial inhomogeneities, as well as a slightly weaker density on the edge of the switched up-area resulting from the Gaussian spot, it cannot be used to determine the magnitude of these variations across the condensate. Note that in our birefringence condition ($\Theta=19^\circ$), $n_x > n_y$ such that Fig.S5.b-d shows only a small fraction of the total polariton density.

- (a) **As suggested by the referee, let us first examine whether the Gaussian spatial intensity distribution in the spot could account for the measured shape of the dispersion relations, and in particular for the too low speed of sound.** Doing so using the full theory is really hard: the specific shape of the cloud strongly depends on the bistability features and, in turn, on the details of the switch-off jump at the edge of the cloud. However, we can take the simplifying assumptions of the local density approximation (LDA), which is reasonable, if not exact, considering the large size of the Gaussian spot ($\sigma=25\mu\text{m}$ radius).

We thus assume a pure, scalar polariton condensate for which the excitation dispersion relation $\omega_B(k_{\parallel}, \omega, n)$ is given by Eq.(2) of the main text. The pump spot is taken Gaussian: $I_p(r) = I_0/(\pi\sigma^2) \exp(-r^2/\sigma^2)$. The local polariton density $n(r)$ is derived within the LDA from the steady-state GGPE equation:

$$I_p(n) = n \left[(\Delta + gn)^2 + \left(\frac{\gamma}{2}\right)^2 \right].$$

Using the experimental parameters $\hbar\Delta = -0.79$ meV, and $\hbar\gamma = 0.4$ meV (WPA), we obtain the hysteretic plot $I_p(n)$ shown in Fig.R2.a, and the density distribution $n(r)$ (Fig.R2.b). This density profile is in qualitative agreement with the measured one shown in Fig.S5.c,d in terms of its “flatness” in the switched-up area (excluding the fluctuations due to the weak disorder). Assuming a Lorentzian lineshape, the inhomogeneous spectral function $F(k, \omega)$ is obtained in the LDA approximation as

$$F(k_{\parallel}, \omega) = \iint_f dx dy n(x, y) \mathcal{L}[\omega, \omega_B(k_{\parallel}, \omega, n(x, y)), \gamma],$$

where the factor $n(x, y) dx dy$ is proportional to the local intensity $dI(x, y)$, $\mathcal{L}[\omega, \omega_0, \gamma]$ is a Lorentzian lineshape centered at ω_0 and of linewidth γ , and f is the spatial filter which is centered on the switched-up area, with a smaller diameter in order to reject the switching front separating the two regions (red dashed line in Fig.R2.b). The calculated spectral function is shown in Fig.R2.c in colorscale. Like in the manuscript, it is compared with the two limiting cases: *homogeneous* pure polaritons condensate dispersion relation (cyan dashed line), and rigid blueshift (black dashed line), both with a blueshift equal to the experimental one (0.85 meV, WPA.2). We see very clearly that in spite of the inhomogeneity from the Gaussian spot, the fit with the pure condensate limit is nearly perfect.

This is in contradiction with the measurement, in which the speed of sound is found twice too low as compared to the homogeneous pure condensate limit. This is far too large a deviation, even in the local density approximation.

Note that the LDA being an approximation, we tested parameters slightly different from the experimental ones in terms of linewidth and laser detuning Δ , for which the switched up density distribution is less flat. The obtained deviation between the homogeneous and the inhomogeneous model remains also largely negligible.

- (b) This analysis provides another relevant insight. In spite of using a Gaussian spot for the excitations, **we see that the hysteretic behavior offers a “flattening” of the polaritons density**, which limits the inhomogeneous broadening of the excitations. This is also true in the case where a reservoir is involved. In essence, the nonlinearity acts as a spatial “tophat” filter for the Gaussian pump, a strategy often used in optical spectroscopy to mimic a quasi-homogeneous spot

Fig.R2 (new Fig.S7) – a) calculated $n(I_0)$ for $\hbar\Delta = -0.79$ meV, and $\hbar\gamma = 0.4$ meV from the homogeneous steady-state GGPE. b) calculated Profile of the polariton density $n(r)$ in the LDA, for a Gaussian spot of size $\sigma=25\mu\text{m}$ and a peak intensity I_0 close enough to the switch down point in order to get close to the experimental blueshift (of WPA.3) $\hbar\omega_{BS} = \hbar\gamma n = 0.85$ meV. The dashed red line show the spatial filter used in the experimental and in the calculation. c) calculated inhomogeneous spectral function $F(k_{\parallel}, \omega)$ in the LDA. The cyan dashed line is the theoretical excitations dispersion relation of the homogeneous pure condensate. The black dashed line is the homogeneous rigid blueshift. d) cross-sections of $F(k_{\parallel} = k_j, \omega)$ for $k_j = \{0, 0.6, 1\} \mu\text{m}^{-1}$ in red blue and green respectively. The dashed line show the homogeneous lineshape $\mathcal{L}[\omega, \omega_0, \gamma]$ for comparison.

intensity distribution. The positive influence of this flattening effect is visible (within the LDA) in Fig.R2.d, where inhomogeneous spectra $F(k_{\parallel} = k_j, \omega)$ for three values of k_j are plotted (color lines) and compared with the homogeneous lineshape $\mathcal{L}[\omega, \omega_0, \gamma]$ (black dashed line). In spite of the Gaussian excitation, we see a vanishing deviation between the two. This is in agreement with the experimentally measured linewidth shown in Fig.S6. Indeed, if we look at the excitation spectrum linewidth at $\theta=0$, for which the effect of the spatial filter is minimum (due to vanishing group velocity), we see that in spite of a total blueshift of 0.85meV (Fig.6.a red line, $\theta=0$), the measured linewidth of 0.4meV remains unchanged as compared to that in the non-interacting regime (Fig.6.a black line, $\theta=0$), within the experimental uncertainty. This is a solid indication that, thanks to the nonlinear “flattening” of the density, the Gaussian spot shape has a negligible influence on the excitations lineshape and on the dispersion relation shape.

Fig.R3 (new Fig.S8): Sections of the spectral functions $I(\theta, \omega=\omega_j)$ as a function of the angle (linked to the wavevector k) for different values of ω_c , increasing from a) to f). Solid line: measurement; dashed line: theoretical model of the spectral function.

- (c) **As mentioned above, the darker and brighter patches observed in Fig.S5.c,d are attributable to a local disorder of weak amplitude (as compared to the linewidth) and correlation length in the 3-5 μm (cf. Fig.2.b).** Again, they are of unknown relative amplitude with respect to the total condensate density. The LDA approximation is probably not good enough to analyze their influence on the excitation dispersion relation. However, a disruptive disorder should result in (i) a significant broadening and asymmetry of the measure spectra $I(\theta = \theta_j, \omega)$, and (ii) on the signatures of localization: spectrally narrow states, broad in momentum space, and flat in dispersion.

We have looked for both signatures in the experimental data and found none of them. Measured spectra $I(\theta = \theta_j, \omega)$ are shown in detail in Fig.S2.b, and exhibit no sharp peaks indicative of localization within the experimental uncertainty. We have also looked at cross-sections at fixed energies $I(\theta, \omega = \omega_j)$ (red lines in Fig.R3), and compare them with the spectral function of the free-polariton dispersion $F_p(\theta, \omega = \omega_j)$ (dashed blue lines in Fig.R3). While this is obviously an approximate model, it is sufficient to check whether some anomalous broadening in momentum space are present in the experimental peaks. The linewidths used in this simple comparison are extrapolated from the table in section II of the SI. Very small corrections to the cross-section energy are applied in order to match angular-peaks from the one-body model and the experiment. It is clear from this comparison that we do not see any momentum broadening or signatures of localization.

➔ **This discussion is given in detail in two new subsection V.3 and V.4 in the SI, and two sentences have been added in the main text**

2. *Note that the linear dispersion is shown for angles $<5^\circ$, while it is almost parabolic for angles $>5^\circ$. At the same time, it is particularly difficult to estimate the exact dispersion for angles $<5^\circ$ from the condensate due to the energy filtering. As shown in Fig. S2, for angles $<4.7^\circ$ the peak of the Lorentzian is masked, and we have to rely only on the Lorentzian fitting. If multiple resonances are present below 5° , it would be difficult to distinguish between them.*

The referee is correct that at low angles we don't see the whole peak of the collective excitation resonance. In fact in the range $\theta < |3^\circ|$ the maximum of the peak is missing. In this range however, the agreement between an exact Lorentzian and the data points, consisting in the blue and red side of the peak, is of $(1-R^2)=98.5\%$, which correspond to $\pm 11\mu\text{eV}$ accuracy of the central peak position.

This accuracy does not allow distinguishing the double peak structure consisting of a sum of two close Lorentzians as predicted by the theory. However, the good quality of the peak flanks data, essentially rule out the possibility of a strong participation of localized state that would show up as narrow non-dispersed multiple peaks.

3. *The spatial filtering over smaller regions will probably limit irremediably the resolution in the Fourier plane, but still a discussion about the way one can exclude this possibility seems required.*

Our aperture-like filtering indeed reduces the momentum resolution giving an angular spread of $\delta\theta \approx 1.5^\circ$ in both WPA and WPB. While this effect is indeed present, and contributes in broadening the linewidth at larger angles as shown in Fig.S6, it is not hiding the interesting features of the dispersion that take place within a window of $\Delta\theta \approx 5^\circ$ as can be seen in the DRs in fig.3. Also, as mentioned in pt.2 above, the corresponding energy window goes sufficiently far beyond (above) the energy-filtered region.

➔ **We have added a mention of the filter-limited angular resolution in the main text**

4. *The interpretation of the results is slightly complicated by the cross-polarized experiments. The birefringence is useful to filter out the scattered laser but a more sophisticated vectorial model is needed. The presence of multiple branches and the partial optical access makes less clear which branches are populated. This should be explained more extendedly and probably a plot of the multiple branches could help to follow the description.*

➔ **We now show and discuss the 5 theoretical branches in different excitation conditions in the SI, (section VIII, and new Fig.S10)**

Minor points

- *Is the healing length also reduced? would it be possible to measure the healing length and compare it with the speed of sound?*

This is a very interesting question which would deserve a fully dedicated study. Indeed even without the added complication of the reservoir, the healing length is only well-defined in the sonic regime, not in every other states of the upper branch of the hysteresis, where a gap is present. Then, both the reservoir and the vectorial degree of freedom of the condensate further complicates the healing length concept. Understanding this feature goes clearly beyond the scope of this work: it would require a separate

investigation both theoretically and experimentally. Thus for instance the healing length could be obtained from measurements of the of vortices diameter (similarly to [D. N. Krizhanovskii et al. Phys. Rev. Lett. **104**, 126402 (2010)]) throughout different states of $I(P)$, in an area of the fluid which is not under the spot, (i.e. where the condensate phase is free). We thus chose not to discuss it in the present work.

- *Which other systems may show the dynamic separation of the reservoir from the excitations in the superfluid phase?*

Except for the case of second sound physics in equilibrium superfluids, in which unlike in polaritons, the reservoir is made up of the same particles as in the superfluid, and both are at thermal equilibrium with each other, we are not aware of any other system showing a reservoir coupled to the superfluid.

➔ **We have added a mention to this system in the revised intro.**

- *Is the dispersion of excitations at high-k blue-shifted from the parabolic dispersion by the total blueshift, only by the polariton part or a combination of the two?*

The blueshift of high-k modes, defined as the modes in which the kinetic energy \gg the interaction energy, is indeed a non-trivial contribution of the two. It is the sum of the total blueshift $g_R n_R + gn$, plus the term that results from the new shape of the dispersion relation, which is fixed only by the polaritons contribution. This latter term amounts to gn , like in the pure polariton condensate case. The total blueshift of these states thus amounts to $g_R n_R + 2gn$.

- *How is the energy of the condensate precisely estimated if it is not the same as the central laser frequency?*

We are in a fully resonant configuration, in which the condensate energy is exactly that of the laser.

- *How the frequency of the excitations is defined in pg.6: "In the regime where excitations of frequencies.."? Is this giving a cutoff wavevector for the crossover from the rigid-blueshift to the decoupled-reservoir-condensate regimes? How much is it? How does it change by changing γ_R ?*

The frequency we refer to are the excitations frequency $\omega_e/2\pi$ with respect to the condensate, which is the energy reference. $\hbar\omega_e$ is thus defined as its photoluminescence energy minus the condensate energy, which is equal to that of the laser. The y-axes in Fig.3.b,d,g,i and Fig.S4.e-g correspond to $\hbar\omega_e$.

As explained in the reply to referee 1 (6th point in the editorial comments), the different regimes mentioned by the referee take place indeed in different frequency range:

- In the limit of static excitations (frequency $\omega_e/2\pi = 0$), both the reservoir polaritonic density contribute in the excitations.
- This reservoir-condensate coupling breaks down when $0 < \omega_e < \gamma_R$. This is a crossover regime where the reservoir contribution is non zero but decrease for increasing frequency.
- When $\omega_e > \gamma_R$, the reservoir, too slow, does not contribute anymore to the excitations.

→ ω_e has been introduced as the notation for the excitation frequency in the main text and in the relevant figure captions. The paragraph discussing the different regimes of excitations has been extended for better clarity.

- *The small Q factor and spatial inhomogeneity are a limiting factor for the maximum precision achievable even in this accurate experimental realization. Can longer lifetime be useful in the measurements of the actual speed of sound?*

We chose on purpose a microcavity with a relatively small Q factor in order to optimize two conflicting requirements: The finite resolution of our spectrometer (of $70\mu\text{eV}$) fixes the minimum blueshift Δ we can work with. A Δ at least 5 times larger (i.e. 0.4meV) than the resolution can be considered a safe zone. On the other hand, as we are trying to work with a large spot in order to approach as much as possible the spatially homogeneous excitation regime, the max available CW laser power is the second limitation. If we decrease the polariton linewidth (increase Q) for a fixed Δ , the required laser power needed to switch to the upper branch of the hysteresis increases: for instance we need 30mW (as measured with a 5% duty-cycle) at the sample surface to do so with $\hbar\gamma=0.4\text{meV}$, and we would need 75mW to do so with $\hbar\gamma=0.1\text{meV}$.

→ We have added a sentence about that in the method section

- *Does this sentence: “Finally, the speed of sound that would be observed at the sonic point can be deduced from ρ , ..., $c/v=0.5$ and $c/v=0.6$ ” refer to WPA and WPB at power 2 and 3?*

In the previous version of the manuscript, the values c_x/v_c and c_y/v_c were referring to WPA. In the corrected version of, we chose to present c_x/v_c for both WPA and WPB instead, as they have slightly different values, that agree with the slight difference in interaction strength between WPA and WPB.

→ The sentence has been rephrased as “the characteristic speed of sounds amount to $c_x/v_c = 0.42 \pm 0.05$ for WPA and $c_x/v_c = 0.56 \pm 0.12$ for WPB”.

→ We have implemented the other editorial comments noted by the referee

REFEREE 3

We thank the referee for his/her positive view of our work in stating that “the understanding of the collective spectrum of such excitations is fundamentally important for our understanding of the physics of exciton-polaritons”, and for having carefully reviewed our manuscript, in particular the theoretical aspects. We append below our answers to the referee’s comments and the implemented corrections.

1. γ_{bk} in Eq (4) describes the rate with which reservoir excitons are converted back into polaritons. Should not in this case γ_{bk} be multiplied by $|\psi|^2$?

We apologize for the confusing discussion of the interconversion of polaritons into reservoir excitons and back. In the complete quantum kinetic equations for the two populations (including all the required $1+n$ bosonic stimulation factors), there are indeed terms where the conversion rate is multiplied by both $|\psi|^2$ and n_R . However, such terms occur in both directions, and eventually cancel out. The only remaining terms are the one present in the revised equations. In the main text we have included γ_{bk} into a (redefined) total γ_R which is the only quantity to which we have experimental access. As the equations are written now, it is straightforward to see that γ_{in} gives a small correction to γ_c in the equation for ψ , while only the total γ_R matters in the equation for the reservoir n_R .

➔ **The notation has been modified to avoid confusion. A more detailed explanation has been added in the SI section I.**

2. *One of the conclusions of the manuscript is that "the spectroscopic measurement of the collective excitations provides a new and accurate way to extract the polariton-polariton interactions contribution to the total interaction energy, and isolate it from the reservoir contribution." Indeed, it seems that the value of g should immediately follow from the analysis presented. Can the authors please provide the estimates on the polariton-polariton interaction strength that follows from their analysis and relate or contrast their findings with known estimates? In particular, what is the value of g_T used in Section C and Supp. Inf VII?*

The comparison between our model and the experimental dispersion relations provides us with an estimate of the relative contributions to the total blueshift of the reservoir ($g_R n_R$) on one hand, and of the condensate on the other hand ($\bar{g}n$). However, in order to derive an actual value of the polariton-polariton interaction constant g_T (with $2\bar{g} = g_T + g_S$), we need to have an absolute estimate of the polariton density n , or of the pump term $|F|^2$. Since we have a solid measurement of the latter in our $I(P)$ measurement, we can indeed derive an estimate for g_T . A difficulty in doing so is that we need to make a simplifying assumption to take care of the Gaussian shape of the spot. This is done by deriving a geometrical factor in the local density approximation. We thus get a value $g_T = 8 \pm 2\mu eV\mu m^2$ in which the error bar accounts both for the choice of the geometrical factor, and for the error bar in the fitting of ρ . This value is well in line with the literature, with the advantage of ruling out for sure the reservoir contribution. However, in this work, the experimental strategy is not optimum to maximize the accuracy

of the determination of g_T , and the error bar remains significant, like in previous works. A more accurate determination of g_T would require a dedicated experiment.

→ **We have added this value and a short discussion on its derivation, its comparison with the literature and its accuracy in the main text. We also added a new section in the SI (section VII) explaining the method for its derivation.**

3. *Have the same values of γ_R , γ_c , γ_{in} been used in all simulations? It would be good to have their values stated and discussed in the main part of the paper. Supp. Inf. gives these as $\gamma_R = \gamma_{in} = 0.015 \text{ meV}$, $\gamma_c = 0.4 \text{ meV}$. However, the experimental decay rates give the ratio of about 8 between the life-time of polaritons and reservoir excitons (58ps and 400 ps). Where does the discrepancy come from?*

We apologize for the varying values of γ_R found in the manuscript and SI. $\gamma_R = 1.6 \text{ } \mu\text{eV}$ should appear everywhere as derived from the photoluminescence decay measurement. The other value was a propagated misprint. **This has been corrected everywhere.** Note that this misprint had no incidence on the conclusions as only the product $g_R n_R = \gamma_{in} g_R n / \gamma_R$ is determined as a free parameter in the fitting procedure, not the individual terms of this product. Note also that γ_R is a purely excitonic feature that does not depend on the cavity properties. $\hbar\gamma_c = 0.4 \text{ meV}$ is fixed by the linewidth measurement and does not change much between WPA and WPB (cf. referee point 6). γ_c and γ_R are thus kept constant in each WPs. As explained above for γ_R , γ_{in} does not need to be determined either for the derivation of the relative contributions to the blueshift and for the shape of the dispersion relation. It is however quantitatively determined when we derive ρ and g_T , and it is different in WPA and WPB. This difference has been properly accounted for in this new version.

The value of 58ps found in the photoluminescence decay shown in section I in the SI, is the instrument-limited fast component of the photoluminescence decay. The instrument consists in a monochromator fitted with a streak-camera. The value $\tau_{res} = 58 \text{ ps}$ results from the high spectral resolution provided by the monochromator, which is needed to separate properly the polariton signal from the neighboring resonant laser. Therefore, any signal from the microcavity much faster than τ_{res} shows up in this measurement as a decay at τ_{res} . This is the case for the microcavity radiative decay rate $\tau_c = 1/\gamma_c$, that we know from its spectral linewidth, to be of 1.6 ps.

→ **$\hbar\gamma_R$ and $\hbar\gamma_c$ are now given in the method sections for both WPs. We have also added a clearer description of this instrumental bias in the SI (section I)**

4. *The main advantage of the presented theory is that it explains factor 2 discrepancy in the measured speed of sound, but to claim quantitative correctness of the model the agreements in other measured characteristics should be shown.*

As we have shown in the answers above our model is fully consistent with the spectral linewidth (and correctly predicts its k -dependence as is shown in the SI, Section V.2) and the photoluminescence decay. Of course, an important outlook of our work is to study other dynamical properties of excitons polaritons, such as vortex and soliton formation, analogue of Hawking effect etc, but this goes clearly beyond the scope of the current work.

REVIEWERS' COMMENTS:

Reviewer #1 (Remarks to the Author):

In the revised version of the manuscript, the authors have significantly improved their presentation and addressed all of the questions raised by myself and the other referees.

However, I have some comments relating to the new parts of the manuscript. In particular, the authors now include an estimate of the polariton-polariton interaction strength both in the main text and in the supplemental material. They then compare the value they have obtained with the previously reported results (Refs. 42, 55, 56). I find this comparison confusing for several reasons. First of all, the results of Refs. 55 & 56 relate to a different quantum well material (and therefore presumably different properties of an exciton?). Secondly, the study of Ref. 42 reports an elevated value of interaction strength, which is explained by the additional contribution from the reservoir at the location of polaritons. Finally, the authors' claim in the concluding section is misleading: the study of Ref. 58 fully eliminates the reservoir due to the "hole-burning" effect in a pulsed regime of the incoherent excitation, therefore a reservoir cannot "have a dramatic impact on the accuracy" of this latter measurement. In my view, the authors should clarify their statements and put their new results in a proper context before the manuscript can proceed to publication.

Reviewer #2 (Remarks to the Author):

The points raised in the previous report were satisfactorily addressed in the authors' reply. The new version of the manuscript is, in my opinion, solid and clear and I can recommend the publication in the present form.

Reviewer #3 (Remarks to the Author):

I am happy with the revisions and clarifications made by the authors and recommend the paper for publication.

Reply to Referee 1 (R1)

R1: I have some comments relating to the new parts of the manuscript. In particular, the authors now include an estimate of the polariton-polariton interaction strength both in the main text and in the supplemental material. They then compare the value they have obtained with the previously reported results (Refs. 42, 55, 56). I find this comparison confusing for several reasons. First of all, the results of Refs. 55 & 56 relate to a different quantum well material (and therefore presumably different properties of an exciton?).

The interaction strength depends crucially on the quantum well parameters. They can be summarized as follows:

[42] In(4%)Ga(96%)As, 10nm thick
[55] In(4%)Ga(96%)As, 17nm thick ([59] in the new version)
[56] In(4%)Ga(96%)As, 10nm thick ([51] in the new version)
[57] GaAs, 7nm thick ([61] in the new version)
[58] GaAs, 7nm thick ([52] in the new version)
[b] GaAs, 7nm ([53] in the new version)
[this work] GaAs, 7nm

Note that In(4%)Ga(96%)As differ from GaAs by the addition of only 4% Indium. This difference changes very little the background dielectric constant and the exciton binding energy. The difference in interaction strength between the two materials can be safely considered as lower than the experimental uncertainty as suggested also in [58]. The interaction strength depends also on the quantum well thickness (maybe more, considering the relative variations from one ref to the next) that vary from sample to sample.

Our point in this new discussion was not to provide an accurate comparison between the different results in the literature, but rather to do so assuming a large degree of approximation. We tried to convey this precaution throughout the expression “we thus find (...) which mostly in-line with the values previously reported in the literature”.

But we agree with the referee that it is possible to be more nuanced.

R1: Secondly, the study of Ref. 42 reports an elevated value of interaction strength, which is explained by the additional contribution from the reservoir at the location of polaritons. Finally, the authors' claim in the concluding section is misleading: the study of Ref. 58 fully eliminates the reservoir due to the "hole-burning" effect in a pulsed regime of the incoherent excitation, therefore a reservoir cannot "have a dramatic impact on the accuracy" of this latter measurement.

We agree with the referee on both points.

R1: In my view, the authors should clarify their statements and put their new results in a proper context before the manuscript can proceed to publication.

We did the following changes:

- we replaced "*...which is mostly in-line the values previously reported in the literature [42, 55, 56].*"
with

"...which is mostly in-line with the values previously reported in the literature for microcavities of similar or closely related design [56,58,b]."

Where refs [58] and [b] involve QWs identical to ours, and where the reported interaction strength values are in-line with ours : ($g_X=6\mu\text{eV}\cdot\mu\text{m}^2$ and $g_X=[2,9] \mu\text{eV}\cdot\mu\text{m}^2$ respectively). And [56] has the smallest difference with respect to our system design: In(4%)Ga(96%)As instead of GaAs QWs, with a thickness of 10nm instead of 7nm.

- We also replaced

"Along the same lines, the presence of this reservoir may have a dramatic impact on the accuracy of recent measurements of the polariton-polariton interaction constant [57, 58]."

with

"Along the same lines, the presence of an incoherent excitonic reservoir is a convincing argument [a] to explain the overestimated polariton-polariton interaction in [57]."

- New references:

[a] M. Pieczarka, M. Boozarjmehr, E. Estrecho, Y. Yoon, M. Steger, K. West, L. N. Pfeiffer, K. A. Nelson, D. W. Snoke, A. G. Truscott, E. A. Ostrovskaya, Effect of optically-induced potential on the energy of trapped exciton-polaritons below the condensation threshold, arXiv:1808.00749

[b] Lydie Ferrier, Esther Wertz, Robert Johne, Dmitry D. Solnyshkov, Pascale Senellart, Isabelle Sagnes, Aristide Lemaître, Guillaume Malpuech, and Jacqueline Bloch, Interactions in Confined Polariton Condensates, Phys. Rev. Lett. 106, 126401 (2011).